# *Sphenophorus levis* Behavior Studies: Evaluating Insect Attractiveness or Repellency to One Insecticide Treatment and Assessing Nocturnal Insect Activity and Location Pattern

**DOI:** 10.3390/insects14020205

**Published:** 2023-02-18

**Authors:** Pedro Henrique Urach Ferreira, Marcelo da Costa Ferreira

**Affiliations:** 1Pedro Henrique Urach Ferreira, Science Production Department, São Paulo State University, Jaboticabal 14884-900, SP, Brazil; 2Agricultural Science Production Department, São Paulo State University, Jaboticabal 14884-900, SP, Brazil

**Keywords:** sugarcane weevil, behavior, soil, circadian rhythm

## Abstract

**Simple Summary:**

The sugarcane weevil *Sphenophorus levis* is a soil pest that causes extensive damage in sugarcane. The association of poor insect control by available pest management tools with the current incomprehension of the insect’s behavior favors the expansion of *S. levis* infestations to several sugarcane farming regions. In this study, we evaluated the behavioral aspect of *S. levis* in terms of reaction to one recommended insecticide dose to assess if its application would either attract or repel *S. levis* adults. We also evaluated the hourly activity and location pattern of *S. levis* adult insects through the day by simulating their natural habitat with sugarcane plants and soil. The results indicate that *S. levis* adults were not attracted nor repelled by the recommended insecticide dose; therefore, its behavior should not lead to any positive or negative effects for the tested insecticide dose performance. *S. levis* activity results showed that insects are primarily nocturnal with most activities (walking, digging and mating) starting and finishing at night. Most insects remained hidden under the soil surface during the day and night; however, insects were exposed (out of the soil) up to four times more during the evening than throughout the day, suggesting that nocturnal insecticide applications could potentially improve *S. levis* control.

**Abstract:**

*Sphenophorus levis* Vaurie, 1978 (Coleoptera: Curculionidae). is a difficult to control pest in sugarcane that causes great damage to the subterranean part of the plant. Low insect control is the result of the pesticide application technology adopted but also a consequence of the lack of studies regarding the pest’s behavior. This research aimed to examine the attractiveness and repellency of one labelled insecticide dose to *S. levis* adults and to evaluate the activity and location behavior of *S. levis* adults under hourly observations over 24 h. Repellency and attractiveness studies were conducted in free-choice tests with treated soil with an insecticide product composed of lambda-cyhalothrin + thiamethoxam active ingredients and untreated soil. Insect activity and location behavior studies were assessed by conducting hourly observations of *S. levis* adults in containers with soil and sugarcane plant. The results indicate that *S. levis* adults are not repelled nor attracted to soil treated with the labelled dose of lambda-cyhalothrin + thiamethoxam in sugarcane. Additionally, insects presented nocturnal behavior for most activities (walking, digging and mating) starting at 6:00 p.m. until 2:00 a.m. An average of 21% of insects were out of the soil at night while the majority, 79%, remained inside the soil. During the day, most insects, 95%, remained hidden in the soil. Exposed insects were primarily located on the soil surface. According to these results, nocturnal insecticide applications may improve *S. levis* adult control due to greater insect activity and exposure at night.

## 1. Introduction

*Sphenophorus levis* Vaurie, 1978 (Coleoptera: Curculionidae), commonly known as the sugarcane weevil, is an important soil pest in sugarcane (*Saccharum officinarum* L.) causing significant negative impacts in the sugarcane industry. The presence of this pest has increased in Brazil over the last twenty years following a shift from the manual burnt sugarcane harvesting system to mechanical sugarcane harvesting without burning. As a consequence of not using fire as a tool to facilitate harvesting, the incidence of pests such as root spittlebug [*Mahanarva fimbriolatta* Stål (Hemiptera: Cercopidae)], sugarcane borer [*Diatraea saccharalis* Fabricius (Lepidoptera: Crambidae)] and *S. levis* increased [1].

As a pest that damages the subterranean part of sugarcane, mostly the rhizomes, *S. levis* usually is found underneath the soil, which makes pest control extremely difficult. Among the pest control options currently available for *S. levis* management, chemical control is most frequently used despite its low efficacy. Some authors have reported low insecticide efficacy for a range of products and field conditions [2,3,4]. Several factors may contribute to low efficacy, including the pesticide application technologies used, active ingredients, environmental and meteorological conditions and insect behavior. 

The interaction of the insecticide’s active ingredient and insect behavior is one example of how pest control may vary. Some authors have reported, for instance, the repellency of insecticides to certain insects, including the repellency of pyrethroids and neonicotinoids to Mexican bean beetle [*Epilachna varivestis* Mulsant (Coleoptera: Coccinellidae)] and the repellency of imidacloprid to pollinator beetles [5,6]. Hence, if *S. levis* is repelled by the insecticides applied in sugarcane, there is a chance of reduced pest control. However, no studies have yet investigated the repellency or attraction of insecticides to *S. levis*. In fact, one author has already stated the necessity of determining the repellency potential of protection products to *S. levis* when studying the control efficacy of *Beauveria bassiana* (Bals.) Vuill. to *S. levis* [7].

In addition to examining the repellency potential of applied substances for pest control, the understanding of insect behavior is vital for an effective integrated pest management program. Regarding the behavior of *S. levis*, some authors have reported, for example, on the gregarious activity of insects that tend to aggregate in distinct points distributed in the field [8,9,10]. In a study observing the spatial distribution of *S. levis*, the range of adult incidence in the field varied from 28 m to 53 m [9]. Another described behavior of *S. levis* is the low flight capacity of adults, similar to other Curculionidae and *Sphenophorus* species; this characteristic was also reported for hunting billbugs [*Sphenophorus venatus vestitus* Chittenden (Coleoptera: Curculionidae)] with limited flight ability [11,12,13]. Some reports have also indicated that *S. levis* adults have a nocturnal behavior even though no previous studies have examined insect activity [14]. Although several Curculionidae species are known for their nocturnal behavior, such as the banana weevil [*Cosmopolites sordidus* (Germar) (Coleoptera: Curculionidae)] with peak activity hours ranging from 21:00 to 4:00 a.m. and *S. venatus vestitus* being most active from 00:00 to 4:00, no research has been conducted to evaluate the peak activity hours of *S. levis* [15,16]. Pest control methods can, therefore, be enhanced by investigating times during which *S. levis* adults are most active and unprotected. 

Due to the importance of better understanding *S. levis* behavior, including its perception, repellency or attraction to applied insecticides and its activity throughout the day, the objectives of this study include assessing the repellency or attractiveness of *S. levis* adults to one insecticide labelled dose for sugarcane application, and evaluating the activity and location of observable *S. levis* adults. Therefore, the present study investigated the hypothesis of *S. levis* adults being attracted or repelled in relation to one insecticide treatment and we examined the hypothesis that *S. levis* adults are more active and exposed (not in the soil) at night compared to during the day.

## 2. Materials and Methods

### 2.1. Experiment 1—Insecticide Repellent/Attractive Activity

A laboratory experiment studying the repellency of one insecticide at the labeled rate in sugarcane on *S. levis* adults was conducted in 2021 in Jaboticabal, SP, Brazil. The experiment was conducted in a randomized complete block design with four replications and was performed in duplicate with first and second experiment replicates occurring on 4 June and 15 June, respectively. 

A structure of five circular plastic containers, 1 L, with one central container (E) connected by plastic cylindrical hose outlets, measured as having a 9.53 mm diameter and 10 cm length, to the other four containers (A, B, C, D), was built (Figure 1c) based on previous studies [17,18,19]. Containers A and B (controls) were arranged diagonally and were filled with 80 g of untreated soil. Containers C and D were filled with 80 g of soil treated with insecticide (Figure 1a). Insecticide soil treatment was conducted in 50 L pots with ratoon sugarcane plants of the CTC 4 variety (Centro de Tecnologia Canavieira S.A., Piracicaba, SP, Brazil) in a pot mixture of soil, sand and manure in a proportion of 3:1:1, respectively. Soil samples were taken for soil analysis at the Soil Fertility Laboratory at UNESP following the methodology of Raij et al. [20], to determine organic matter content (14 g dm^−3^) cation exchange capacity (73 mmolc dm^−3^), base saturation (81%) and soil pH (6.0).

Insecticide treatment in soil consisted of liquid applications in four pots with one commonly used commercial insecticide mixture of lambda-cyhalothrin + thiamethoxam for sugarcane applications (Engeo Pleno™ S, Syngenta, Basel, Switzerland) at the recommended dose of 2 L ha^−1^, representing 212 g a.i. ha^−1^ of lambda-cyhalothrin and 282 g a.i. ha^−1^ of thiamethoxam, with an application volume of 200 L ha^−1^. A 10 mL syringe was used to simulate the stream jet nozzles used in ratoon field applications. Four untreated pots were also included. 

A uniform amount of soil (160 g) was collected from the 5 cm surface of each treated and untreated pot, placed in individual plastic bags and distributed in each circular container. In containers A, B, C and D, one sugarcane stalk was cut in half, weighing 35 g and 10 cm long (Figure 1a), and was placed on top with the cut surface facing the soil. The sugarcane stalk was of the CTC 4 variety and was collected from sugarcane fields at the Research Experimental Station, UNESP, Jaboticabal, SP, Brazil. In central container E, five *S. levis* adults were released (Figure 1b); after 24, 48 and 72 h, the total number of insects per container was assessed (Figure 1d). Insects in containers were maintained in controlled conditions in the laboratory under a 12 h photoperiod, at room temperature (22.3 °C ± 1.4) and relative humidity (59% ± 2). The sex ratio of *S. levis* adults was 1:1. Room temperature and relative humidity were measured with a digital thermo hygrometer Jprolab (JProlab, São José dos Pinhais, PR, Brazil). The original population of *S. levis* adults was collected between March and May, 2021, in sugarcane fields in Jaboticabal, SP, Brazil with previous infestation history and no insecticide application in the year. 

The percentage of repellency (RI) was calculated in accordance with the method described by Mazzonetto and Vendramin [17] and in Viteri Jumbo et al. [19] following Equation (1):(1)$$\mathrm{RI}=\frac{\left(2\times T\right)}{\left(T+C\right)}$$
where RI refers to the repellency index, T represents the percentage of insects in the treated containers and C represents the percentage of insects in the untreated containers. RI values indicating repellency levels ranged from 0 to 2 and results were classified accordingly: when RI < 1 repellence (R) was detected; when RI = 1 neutral activity (N) was detected; when RI > 1 attractivity (A) was detected. To improve RI classification (CL) accuracy, the standard deviation (SD) value of each treatment was considered when classifying repellency. Therefore, each treatment was only considered repellent or attractive when the RI value was outside of the neutral RI value within the SD range (out of 1.00 ± SD).

*S. levis* adult mortality evaluations were not considered in the present experiment as the main objective was to evaluate the repellent or attractive activity of the tested insecticide. A related study examining *S. levis* adult mortality and toxicity levels with insecticides was conducted by Ferreira [21]. 

### 2.2. Experiment 2—Nocturnal Adult Activity Pattern

A laboratory study evaluating the activity and location pattern of *S. levis* adults under semi-controlled conditions was conducted in Jaboticabal, SP, Brazil, in 2021. Three observational experiment replicates were conducted on 11 August, 18 August and 27 October, respectively.

Sugarcane ratoon plants (60 days after harvest) of the CTC 4 variety (Centro de Tecnologia Canavieira S.A., Piracicaba, SP, Brazil) were collected in a field without a history of *S. levis* infestation on 7 July 2021, at the Research Experimental Station, UNESP, Jaboticabal, SP, Brazil. Sugarcane plants were carefully collected while maintaining the rhizome and superficial roots; they were then inspected for any *S. levis* damage or insect presence to ensure plants were not infested with *S. levis* larvae, pupae or adults. On the same date that plants were collected, they were transplanted in 4.5 L square containers (26.6 cm × 26.6 cm × 9 cm) containing 4.1 Kg of soil, sand and manure in a proportion of 3:1:1, respectively (Figure 2a). In the same containers, 35 cm plastic sticks were attached at each container corner and a voile fabric was used to surround it (Figure 2c). 

The *S. levis* adults used in the study originated from sugarcane field collections in March, April and May, 2021, using baits of sugarcane stalks cut in half (30 cm) that were immersed in 50 L water containers with a 10% sugar solution for 24 h following the adapted methodology [22]. Containers with sugarcane plants were kept outside under natural light, temperature and relative humidity to simulate field conditions. During the experiment, in case of rainfall events, containers were covered to ensure similar meteorological conditions, while ensuring the conduction of activity evaluations was feasible. The photoperiod, hourly temperature and relative humidity of each experiment replication were recorded. Temperature and relative humidity were measured with a digital thermo hygrometer Jprolab (Jprolab, São José dos Pinhais, PR, Brazil) during hourly insect activity evaluations. Yellow neon acrylic non-toxic paint (Acrilex Tintas Especiais S.A., SP, Brazil) was used on insects to detect insect location at night (Figure 2b). A small mark on the insects’ elytra was made using a paint brush (size 1) (KOTA NO. 1, Kota Indústria e Comercio Ltda, São Paulo, SP, Brazil). Preliminary observations were conducted to ensure the paint used would not affect the behavior of insects. Twenty *S. levis* adults were placed in the center of each sugarcane pot 12 h before activity and location pattern observations for insect acclimation. Four containers (replicates) were used. 

At midnight, activity pattern evaluations started and were conducted every hour over 24 h. Adults were observed for an average of 3 min per container. The location (e.g., soil surface, cane stalk, tiller base, leaf, not visible) and behavior (e.g., walking, digging, mating, inactive) of each *S. levis* adult per container was recorded (Figure 2d). A blacklight lantern WY6548 model (Coquimbo, Shenzhen, China) was used at night to evaluate insects with minimum disturbance. The day after the third experiment replicate was conducted on 27 October, the sugarcane ratoon plants and soil of each container were removed for a visual assessment of the number and location of remaining *S. levis* per container. Experiments 1 and 2, and the collection of sugarcane plant material and insects in the field were all conducted following relevant institutional, national and international guidelines and legislation.

### 2.3. Data Analysis

A statistical analysis for both experiments was conducted using the R Studio Version 1.4.1717 software [23]. In the first experiment studying insect repellency, the evaluation period, experiment date and treatment were treated as independent variables and the percentage of insects per container was treated as a dependent variable in a general linear model with a quasi-binomial distribution. For both studies, goodness of fit of models were assessed using half-normal plots with simulation envelopes and the hnp package [24] in R software and based on the Akaike information criterion (AIC) and residual deviance values. Insect repellency results were submitted for an analysis of deviance (type II Wald chi-square tests) and significant differences between treatments were analyzed using the emmeans package [25] with Sidak’s test at *p* < 0.05. 

In the second study, the number of observable *S. levis* adults was treated as the dependent variable and an hour of evaluation was treated as an independent variable. Container repetition was treated as a random effect. A generalized linear mixed model was adopted using the glmmtmb package [26] with a Poisson distribution. The mean number of observed insects was submitted to an analysis of deviance (type II Wald chi-square tests) using the Car package [27] and significant differences between hours of evaluation were analyzed using the emmeans package [25] with Sidak’s test at *p* < 0.05. Pearson correlations between the dependent variable of visible insects per container and both temperature and relative humidity variables observed at each hour were assessed with the correlation function in R software.

## 3. Results

### 3.1. Experiment 1—Insecticide Repellent/Attractive Activity

There were no significant differences of *S. levis* repellency regardless of experiment date (*p* = 0.988), evaluation period (*p* = 0.999) and treatment (*p* = 0.728). Therefore, no insect preference was observed between soil treated with lambda-cyhalothrin + thiamethoxam and untreated soil, regardless of the evaluation period (24, 48 and 72 h after insect exposure) and experiment date (4 June and 15 June) for the recommended product dose tested. At the first evaluation, at 24 h, 50% of insects had moved to untreated soil and 50% to treated soil, representing an RI value of 1 and repellency classification of neutral activity (N), as shown in Figure 3. After 48 h, 56% of *S. levis* adults were placed in containers with untreated soil, with an RI value of 0.88 and neutral activity classification (Figure 3). At 72 h, 52% of insects moved to containers with treated soil, representing an RI of 1.04 and neutral activity classification (Figure 3).

### 3.2. Experiment 2—Nocturnal Adult Activity Pattern

Insect activity was significantly affected by time during the 24 h period for the experiments on 11 August (*p* = 0.0147) and 17 August (*p* < 0.0001), as shown in Figure 4, Figure 5, Figure 6, Figure 7 and Figure 8. The experiment on 27 October showed no significant effect of time on insect activity (*p* = 0.0527), as shown in Figure 6 and Figure 9. *S. levis* adults were more active during the night for all experiments. On 11 August, most insects started to become exposed as they left the soil at 7:00 p.m. until 6:00 a.m. and were mostly hidden in the soil from 7:00 a.m. until 6:00 p.m. During the activity peak at night, insects were either resting, walking, digging or mating (Figure 4, Figure 5 and Figure 6) and were mostly located on the soil surface and subsurface (Figure 7, Figure 8 and Figure 9). Mating was only observed at 8:00 p.m. on 11 August. Despite the activity peak at night, most insects were hidden underneath the soil. During the most active period, at 00:00, 21% of insects were exposed while 79% were hidden for that day. During the day, most insects were hidden, with an average of 98% of adults hiding. There was a weak positive correlation (r = 0.22) between exposed insects and relative humidity and there was a weak negative correlation (r = −0.20) between exposed insects and air temperature on 11 August. On 11 August, sunrise occurred at 06:39 a.m. and sunset was at 5:57 p.m., with a total day length of 11 h and 17 min. 

On 17 August, insect activity was mostly observed from 7:00 p.m. until 11:00 p.m. with one exception at 11:00 a.m., during which insect walking was also observed (Figure 5). Mating behavior was observed at 5:00 p.m. and 7:00 p.m. Most exposed/visible *S. levis* adults were either on the soil surface or subsurface (Figure 7, Figure 8 and Figure 9). Most insects were also hidden underneath the soil even at highly active periods. At 8:00 p.m., the most active period on 17 August, 25% insects were exposed/visible with 75% hidden. During the day, an average of 91% of *S. levis* adults were hidden. A very weak negative correlation (r = −0.02) was observed between the number of exposed insects and relative humidity while a very weak positive correlation (r = 0.03) was seen for the number of exposed insects and air temperature on 17 August. The sunrise of 17 August was at 06:35 a.m. and sunset was at 5:59 p.m. (11 h 23 min day length).

On 27 October, insect activity was mostly observed from 2:00 p.m. until 00:00 but the maximum number of visible insects was observed at 6:00 and 7:00 p.m. Mating was observed at 7:00 p.m., 8:00 p.m. and 00:00. Most insects were resting on the soil surface/subsurface or on sugarcane tiller base. At the period of most active insects, 7:00 p.m., 16% of adults were exposed while 84% were hidden. During the day, an average of 96% *S. levis* adults were hidden underneath the soil. There was a weak positive correlation (r = 0.10) between exposed insects and relative humidity and there was a weak negative correlation (r = −0.12) between exposed insects and air temperature on 27 October. The sunrise on 27 October was at 05:34 a.m. and sunset was at 6:20 p.m. (12 h 46 min day length) and two rainfall events occurred on that day. The first rainfall event occurred from 4:30 p.m. to 5:40 p.m. and the second rainfall event started at 9:30 p.m. and lasted until 11:50 p.m. 

Despite *S. levis* adults being more active at night, most adults were hidden underneath the soil surface. On average, considering all three experiment dates, 21% of insects were exposed at night while 79% were hidden inside the soil at night.

During sugarcane removal following the day of experiment 3 at 11:30 a.m., when assessing insect number and location, it was noticed that all insects were located underneath the soil, with 92% of adults attached to sugarcane rhizomes and roots (Figure 10, Appendix A) and 8% were freely in the soil (Figure 11). 

## 4. Discussion

### 4.1. Experiment 1—Insecticide Repellent/Attractive Activity

*Sphenophorus levis* adults were not attracted nor repelled by treated soil with lambda-cyhalothrin + thiamethoxam at the recommended dose for sugarcane applications. According to the classification of Mazzonetto and Vendramin [17], this result of no attraction and no repellency can be classified as neutral activity. Even though some studies indicated pyrethroid and neonicotinoid insecticides having a repellent activity (e.g., pyrethroid repellency to *E.* varivestis, Dobrin and Hammond [5], and imidacloprid repellency to different species of pollinator beetles, Easton and Goulson [6]), both active ingredients used in the present study were not repellent to *S. levis* at the recommended dose in sugarcane applications. Even though the present study considered one insecticide at the labelled dose which is commonly used by sugarcane farmers, future studies could also include different insecticide concentrations and IPM strategies. When studying the repellency of clove and cinnamon essential oils on bean weevil (*Acanthoscelides obtectus*), for example, it was observed that *A. obtectus* were only repelled by higher dosages of cinnamon oil, specifically those above the lethal dose of 50% (LD_50_) [19]. In addition, despite the few insecticide options currently available and used for *S. levis* control in sugarcane, other potential insecticides should also be tested for repellency against *S. levis* in future studies. 

In practical terms, no insecticide repellency to *S. levis* may be positive if proper pest control is achieved once *S. levis* adults are attracted to sugarcane plants and come into contact with the product. However, optimal *S. levis* control has not been a reality as several authors have reported [2,3,4]. Moreover, no insecticide repellency to *S. levis* may be necessary when adopting behavioral control methods, including the attract-and-kill baiting approach used in sugarcane. In this method, sugarcane stalks are treated with insecticides and distributed across the field, aiming to attract and control *S. levis* adults. Studying the attractiveness of vinasse, a sugarcane byproduct, to *S. levis* adults, a high level of attraction to sugarcane stalk baits mixed with vinasse was observed [28]. Perhaps treating sugarcane stalks with both vinasse and insecticides could improve the attract-and-kill control method. Additionally, if *S. levis* aggregation pheromones, such as 2-methyl-4-octanol could be identified and synthesized, adding them to sugarcane baits would possibly further increase its attraction to insects [8]. For instance, a similar attraction response was observed when testing sugarcane baits mixed with aggregation pheromones from [*Sphenophorus incurrens* Gyllenhal (Coleoptera: Curculionidae)] to capture adults in field [29].

### 4.2. Experiment 2—Nocturnal Adult Activity Pattern

The results of *S. levis* activity indicate a primary nocturnal behavior of *S. levis* adults. Considering all three experiment dates, most insect activities were observed from 18:00 p.m. to 2:00 a.m. The observed nocturnal behavior is in accordance with results reported by authors in another study, namely the study by Casteliani et al. [14], even though no specific hourly observations of activity and location were adopted in that study [14]. Other Curculionidae species are also known for their nocturnal behavior, such as *C. sordidus* with peak activity from 9:00 p.m. to 4:00 a.m., as well as other *Sphenophorus* species [15]. Similar results were described for *S. venatus vestitus*, which were most active between 00:00 and 4:00 a.m. [16]. In fact, one monitoring option recommended for *S. venatus vestitus* in turfgrass is to scout adults at night [30]. In addition, during preliminary tests of the present study, *S. levis* adults were shown to move away when a light source was present, and move towards dark locations such as the soil, a characteristic of negative phototaxis. In fact, extraretinal photoreceptors such as the Hofbauer–Buchner eyelet, are known to be responsible for light responses and communication with circadian clocks affecting locomotor and activity behaviors in many insects [31,32]. Regarding the locomotion of observable *S. levis* adults, insects were mostly resting while some were walking on soil. Mating was another observed behavior that was mainly noticed at night, and was also described for *S. venatus vestitus*, with the most occurrence from 00:00 to 4:00 a.m. [16]. *S. levis* adults were mainly located underneath the soil but when exposed, were found mostly on the soil surface or subsurface and sometimes were seen on top of sugarcane leaves, stems and the tiller base, as also observed in another study [14]. 

As all three experiments were conducted in August and October, the low rate of adults emerging from soil (<21%) may be explained by the insect’s main distribution in specific months with higher temperatures and moisture. Several authors have reported a greater field distribution of *S. levis* adults between October and November and between February and March, while larvae distribution was primarily observed during June and July [7,11,33,34]. Despite other physiological factors related to optimum temperature, light and humidity for adult development and behavior, it is hypothesized that adults may be more exposed and active in these specific periods within the year as a consequence of water-saturated soils common to months with heavy rainfall. Thus, soil pores occupied by water might force *S. levis* adults to emerge from the soil subsurface. Studying the effect of soil moisture on *S. venatus vestitus*, for example, it has been observed larvae better developed under 20% of total pore space with water [30]. Which also helps understanding the higher *S. levis* distribution of larvae during dryer periods (June–July). Therefore, it is possible that the percentage of exposed and active *S. levis* adults in the present study were to be higher if conducted during the rainfall season and under high soil moisture. Further studies should be conducted in different periods throughout the year for a better understanding of *S. levis* adult exposure and activity. Additionally, as no strong correlation was noticed for both air temperature and relative humidity in relation to the number of *S. levis* adults exposed, it is assumed that other environmental stimuli, such as soil moisture, are associated with adult behavior and should also be considered in future studies. In contrast, southern corn billbug [*Sphenophorus callosus* Olivier (Coleoptera: Curculionidae)] activity was more closely associated with air temperature than soil temperature and insects were more active during the day, from 12:00 p.m. to 2:00 p.m. [35].

During daylight, the low movement of *S. levis* adults out of the soil and their activity was even more significant. Considering the three experiment dates, on average, 95% of insects were hidden in soil during the day. As previously discussed, *S. levis* can be classified as negatively phototactic, moving away from the light towards dark locations, specifically the soil. As a result, most *S. levis* adults are located underneath the soil during daylight, usually coinciding with the period of pesticide applications for its control in sugarcane. Although most insecticides registered for *S. levis* control are considered systemic with some residual effect and should provide some pest protection over time, most applications are insufficient and ineffective in the control of *S. levis* [2,3,4]. In addition to the difficulty of pesticide application technology’s correct deposition of the pesticide’s active ingredient in the soil/rhizome, current diurnal applications are possibly missing most exposed *S. levis* adults due to their nocturnal behavior. 

Based on the current results, nocturnal insecticide applications could significantly increase the chance of reaching *S. levis* adults and could possibly contribute to better control of the insect. Despite only 21% of adults, on average, being exposed at night in the present study, the possibility of reaching adults during nocturnal applications is up to 4 times higher than diurnal applications, where only 5% of adults, on average, are exposed. The benefit of pesticide applications at night is reported in a study evaluating the effect of application timing on fall armyworm [*Spodoptera frugiperda* Smith (Lepidoptera: Noctuidae)] control with most effective applications conducted at 8:00 p.m., 00:00 and 4:00 a.m. [36]. Another study described similar benefits regarding nocturnal applications, in which the authors observed satisfactory control levels of burrower bug [(*Cyrtomenus mirabilis* (Perty, 1836) (Hemiptera: Cydnidae)] in peanut (*Arachis hypogaea* L.) with different insecticides applied at night [37]. As most insecticide applications targeting *S. levis* are directed towards soil, usually with a full jet nozzle, nocturnal applications should also include one even, flat-fan nozzle for band applications at the plant base to improve spray coverage and deposit on exposed and active *S. levis* adults. For instance, one study compared two application methods for *S. levis* control, using a standard soil application method with one nozzle directed to the soil and another application method with two nozzles where one nozzle sprayed 30% of the application volume in the soil and the second nozzle sprayed 70% of the application volume towards the sugarcane base [38]. According to Dinardo-Miranda [38], the application method with two spray nozzles should be recommended during the rainfall periods due to the greater adult distribution; however, according to the current results, including this application method at night may further improve efficiency. In addition, as previously discussed, during peak populational periods (October/November and February/March) the number of exposed and active *S. levis* adults may probably increase in comparison with these observed results of August and October showing 21% of exposed adults at night. Hence, future studies should evaluate the potential of nocturnal applications of insecticides for *S. levis* control.

Moreover, if future studies show a strong correlation between the behavior of reared/contained *S. levis* adults with the behavior of field *S. levis* adults, a direct monitoring system could be developed for better pest management decisions. Such a system could be used, for example, to monitor contained *S. levis* adults’ activity in real time by providing site-specific information about the period of exposure and, consequently, the best recommended time for insecticide applications targeting exposed adults. In fact, low-cost portable locomotion activity monitor systems have been developed to track field and laboratory insect activity, including circadian rhythm, locomotion and feeding behavior [39]. 

In addition to semi-controlled studies, such as the present behavior experiment, new studies under field conditions should also be conducted considering the possibility of distinct and more accurate insect behavior in real field conditions.

Finally, during the sugarcane and insect removal of each container for an insect number and location assessment, it was noticed that all insects were located underneath the soil with 92% of adults attached to sugarcane rhizomes and roots (Figure 10, Appendix A), with 8% of adults freely in the soil (Figure 11). Several authors have reported on the gregarious behavior of *S. levis* [8,9,10]. Such behavior is induced by aggregation pheromones such as 2-methyl-4-octanol [8]. Additionally, *S. levis* are known to have a slow spatial distribution capacity, ranging from 5.2 to 6.6 m month^−1^ in part because of its rare flying behavior but also because of its aggregation activity [10,33]. Regarding some evident sugarcane damage and openings made by adults on plants from each container, most of it was seen below the ground (Figure 10), as described in another study by Casteliani et al. [14], in which the authors observed 90% damage and openings below the soil surface. 

Despite the important findings observed in the present study, further research should be conducted to better elucidate *S. levis* behavior and biology and consequently improve pest control in sugarcane.

## 5. Conclusions

Soil treated with the labeled dose of lambda-cyhalothrin and thiamethoxam were not repellent nor attractive to *S. levis* adults. Insects presented nocturnal behavior as most activities and the highest number of *S. levis* adults out of the soil were observed between 6:00 p.m. and 2:00 a.m. Despite the nocturnal behavior, most insects remained inside the soil (79%) at night while other insects were either active or inactive on the soil surface, subsurface, plant base and cane stem (21%). During the day, the vast majority of *S. levis* adults were found underneath the soil (95%) aggregating near or attached to the sugarcane rhizome. Based on these results, nocturnal applications of insecticides may improve *S. levis* control in sugarcane.

## Figures and Tables

**Figure 1 insects-14-00205-f001:**
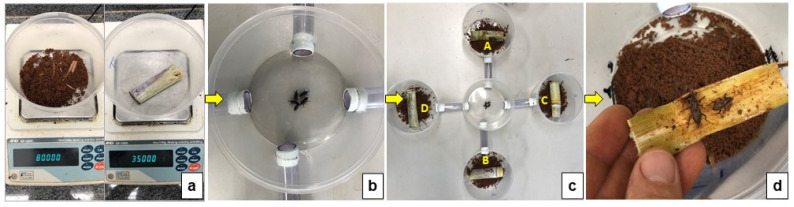
Methodology steps for the insecticide repellency activity including treated soil and sugarcane placement (**a**) in containers; placement of *S. levis* adults in the central container (**b**); study apparatus with five containers; (**c**) repellency evaluations after 24, 48 and 72 h (**d**).

**Figure 2 insects-14-00205-f002:**
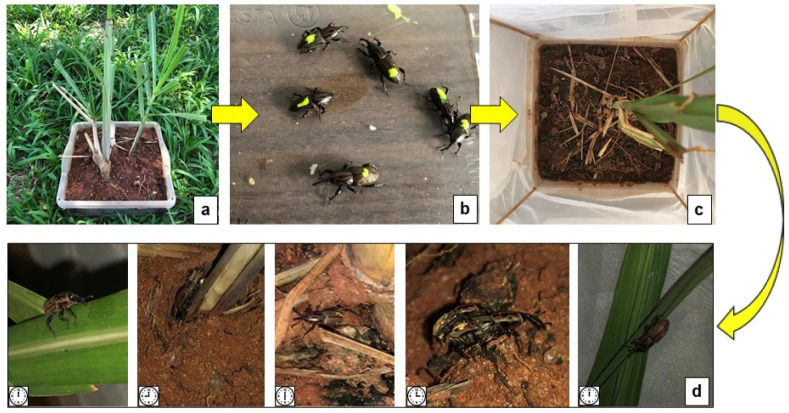
Methodology for the *S. levis* adult activity pattern study including ratoon sugarcane in containers (**a**); placement of 20 marked *S. levis* adults (**b**) in containers under natural weather conditions (**c**); and hourly insect activity evaluations recording adult location and behavior (**d**).

**Figure 3 insects-14-00205-f003:**
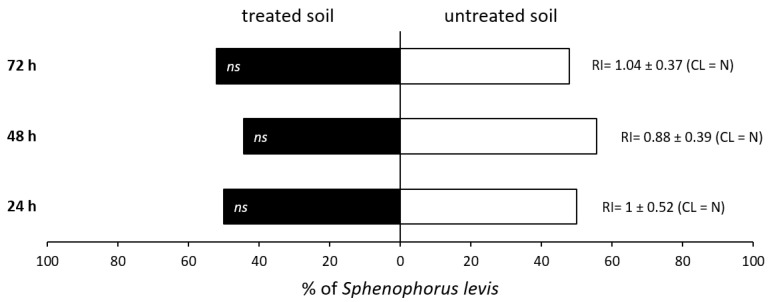
Percentage of *S. levis* adults that moved to soil treated with lambda-cyhalothrin + thiamethoxam and to untreated soil per evaluation period. Each bar represents the mean results of four replicates of two study repetitions. Repellency index (RI) with SD and RI classification (CL) are provided. *ns*—no significant differences were observed at α = 0.05. N—neutral activity.

**Figure 4 insects-14-00205-f004:**
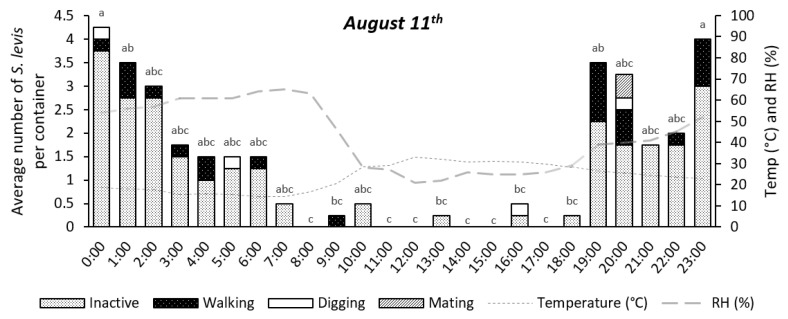
Number of visible *S. levis* adults per container at every hour on 11 August considering insect activity, temperature (temp) and relative humidity (RH%). Bars with mean values followed by same letter are not different at α = 0.05.

**Figure 5 insects-14-00205-f005:**
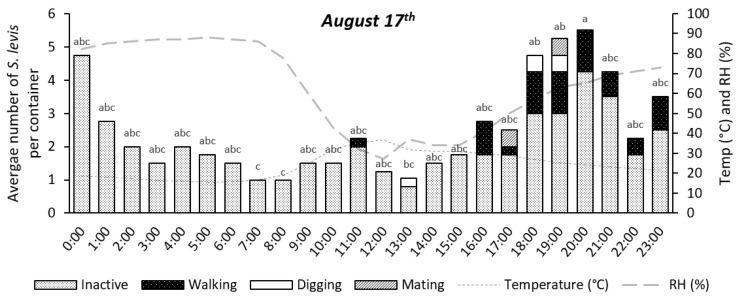
Number of visible *S. levis* adults per container at every hour on 17 August considering insect activity, temperature (temp) and relative humidity (RH%). Bars with mean values followed by same letter are not different at α = 0.05.

**Figure 6 insects-14-00205-f006:**
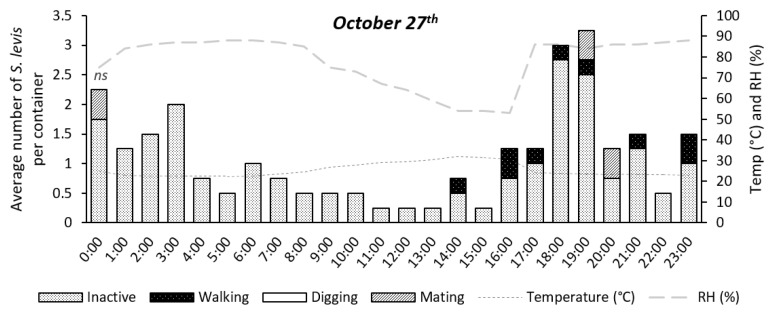
Number of visible *S. levis* adults per container at every hour on 27 October considering insect activity, temperature (temp) and relative humidity (RH%). *ns*—no significant differences were observed at α = 0.05.

**Figure 7 insects-14-00205-f007:**
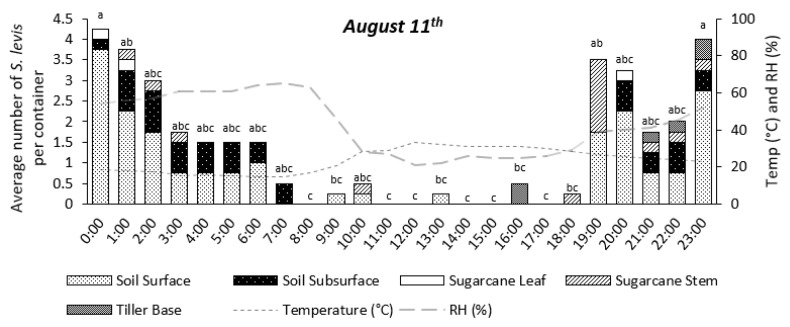
Number of visible *S. levis* adults per container at every hour on 11 August considering insect location, temperature (temp) and relative humidity (RH%). Bars with mean values followed by same letter are not different at α = 0.05.

**Figure 8 insects-14-00205-f008:**
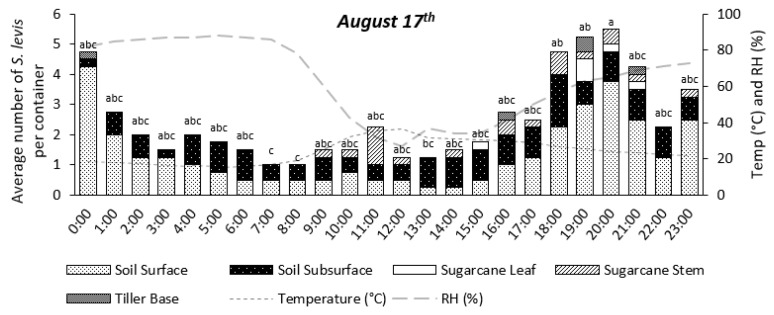
Number of visible *S. levis* adults per container at every hour on 17 August considering insect location, temperature (temp) and relative humidity (RH%). Bars with mean values followed by same letter are not different at α = 0.05.

**Figure 9 insects-14-00205-f009:**
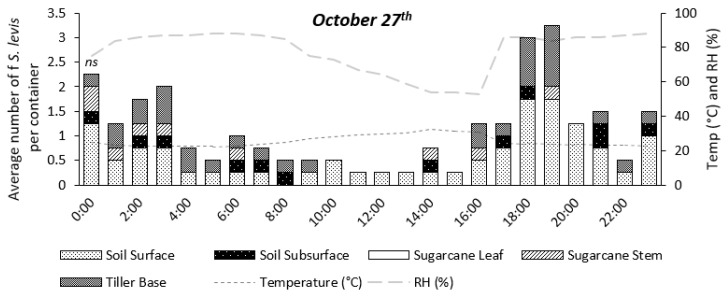
Number of visible *S. levis* adults per container at every hour on 27 October considering insect location, temperature (temp) and relative humidity (RH%). *ns*—no significant differences were observed at α = 0.05.

**Figure 10 insects-14-00205-f010:**
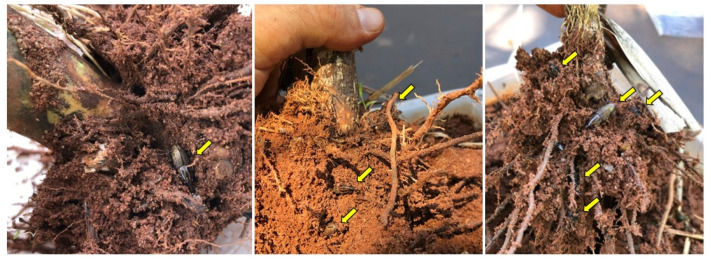
*Sphenophorus levis* adults attached to sugarcane rhizomes and roots. Yellow arrows indicate *S. levis* adults.

**Figure 11 insects-14-00205-f011:**
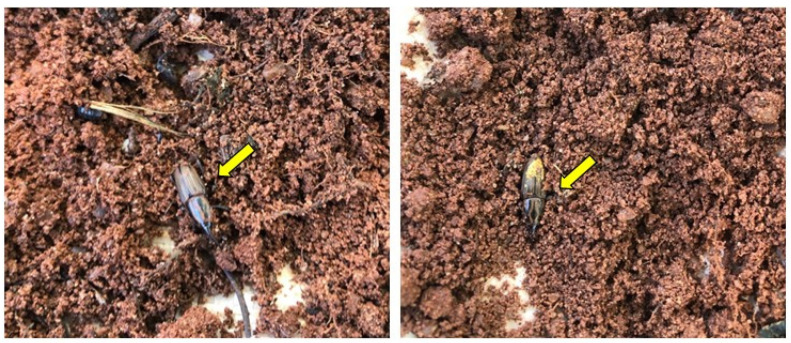
*Sphenophorus levis* adults found freely on soil and not attached to sugarcane rhizomes and roots as indicated by yellow arrows.

## Data Availability

Additional information is available from the corresponding author, P.H.U.F., upon reasonable request, including the datasets generated and/or analyzed during the study.

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
