# Peer review of "Sphenophorus levis Behavior Studies: Evaluating Insect Attractiveness or Repellency to One Insecticide Treatment and Assessing Nocturnal Insect Activity and Location Pattern"

_insects, 2023, doi:10.3390/insects14020205_

Round 1

Reviewer 1 Report

The article "Sphenophorus levis behavior studies: evaluating insect attractiveness or repellency to one insecticide treatment and assessing nocturnal insect activity and location pattern" presents novelty and merity to be published in the current journal. However, I have some suggestions to improve it.

Simple Summary and Abstract: No comments.

Introduction: The hypothesis about the study is missing at the end of introduction.

Material and Methods:

The first (Lines 92 - 128) and fourth (150 - 189) paragraphs of this section are poorly organized, it is too long.

It should be broken into 3 or more paragraphs.

It is necessary to make clear whether the research was conducted in the field condition or in the laboratory.

Thus, it seems that it was not used the experimental design "completely randomized design".

I would suggest to the authors double check if they used Randomized Complete Block Design (RCBD).

Data Analysis:

It seems that a generalized linear model was used in the first study, and a generalized linear mixed model to the second one. But it is necessary to describe which distributions have been used in these models.

Results and discussion: No comments.

Reviewer 2 Report

Dear Authors,

Please check the comments and suggestions. It is a very interesting manuscript and with fine tuning, it would be great for readers.

Reviewer 3 Report

The manuscript is well written, with appropriate statistical analysis and results are well presented.

Below some minor points that can be used to improve the manuscript.

Experiments 1 and 2 – What was the age of adults used? Was there a ratio of male/female? Is it possible to determine the activity by sex (e.g., male vs female)?

Experiment 2 – The authors ran the analysis for each date separately. I wonder if they could combine the three different dates into a single analysis.

Line 238 – I am not sure what the authors meant by insects started to be exposed. Also, please correct time using 7 pm instead of 19:00 pm. Other instances also need to be corrected throughout the manuscript.

Figures 4 and 5 – Please rewrite the y-axis for clarification. Maybe something like – Average number of S. levis adults per container

Lines 297-303 – Please rewrite these sentences. They are long and confusing.
